# 4-Hydroxynonenal Immunoreactivity Is Increased in the Frontal Cortex of 5XFAD Transgenic Mice

**DOI:** 10.3390/biomedicines8090326

**Published:** 2020-09-03

**Authors:** Sang-Wook Shin, Dong-Hee Kim, Won Kyung Jeon, Jung-Soo Han

**Affiliations:** 1Department of Biological Science, Konkuk University, 120 Neungdong-ro, Gwangjin-gu, Seoul 05029, Korea; tlstkd0922@gmail.com (S.-W.S.); kdh9579@gmail.com (D.-H.K.); 2Herbal Medicine Research Division, Korea Institute of Oriental Medicine, Daejeon 34054, Korea; 3Convergence Research Center for Diagnosis, Treatment and Care System of Dementia, Korea Institute of Science and Technology, Seoul 02792, Korea

**Keywords:** Alzheimer’s disease, 5XFAD, 4-hydroxy-nonenal, oxidative stress, frontal cortex

## Abstract

Oxidative stress was implicated in the functional impairment of the frontal cortex observed in early Alzheimer’s disease (AD). To elucidate this role in an animal AD model, we assessed cognitive function of 4-month-old five familial AD (5XFAD) transgenic (Tg) mice using a learning strategy-switching task requiring recruitment of the frontal cortex and measuring levels of 4-hydroxy-2-*trans*-nonenal (4-HNE), a marker of oxidative stress, in their frontal cortex. Mice were sequentially trained in cued/response and place/spatial versions of the water maze task for four days each. 5XFAD and non-Tg mice exhibited equal performance in cued/response training. However, 5XFAD mice used spatial search strategy less than non-Tg mice in the spatial/place training. Immunoblot and immunofluorescence staining showed that 4-HNE levels increased in the frontal cortex, but not in the hippocampus and striatum, of 5XFAD mice compared to those in non-Tg mice. We report early cognitive deficits related to the frontal cortex and the frontal cortex’s oxidative damage in 4-month-old 5XFAD mice. These results suggest that 4-month-old 5XFAD mice be a useful animal model for the early diagnosis and management of AD.

## 1. Introduction

Oxidative stress was reported to play a crucial role in the pathogenesis of neurodegenerative disorders, including Alzheimer’s disease (AD) [1,2,3], where it features prominently in early AD and mild cognitive impairment (MCI) [4,5]. Increased 4-hydroxy-2-*trans*-nonenal (4-HNE)-modified protein has been demonstrated in MCI, as well as in both early and late-stage AD [6,7,8]. These studies indicate that cognitive decline in AD is more pronounced with more severe oxidative damage, as indexed by protein-bound 4-HNE [7]. Interestingly, oxidative damage is more pronounced in the frontal cortex than in other cortices in the AD brain [9,10]. Hence, to shed more light on the molecular/cellular processes of early AD pathogenesis and develop therapeutics for AD, an animal AD model with the cognitive and pathological characteristics of early AD is needed.

Transgenic mouse models of AD have been generated to recapitulate the main pathological characteristics, mainly amyloid-β (Aβ) plaques, and cognitive impairments. One of the most broadly used AD animal models is the five familial AD (5XFAD) transgenic (Tg) mouse. 5XFAD mice start developing cerebral Aβ plaques as early as 2 months of age [11]. A study linking early cognitive impairment to the frontal cortex in 5XFAD mice examined the cognitive status of 2-, 4-, and 6-month-old 5XFAD mice with non-Tg control mice using the olfactory H-maze, in which an animal’s performance is a sensitive marker of frontal cortex status. It reported that cognitive deficits, heavy gliosis, and emerging amyloid plaques were apparent in 4-month-old 5XFAD mice [12]. Their results suggest that 5XFAD mice of that age should be useful when studying early AD and assessing frontal brain function in transgenic AD mice. Working memory was also evaluated in 4-month-old 5XFAD mice using the novel object recognition task with varying retention intervals (10 min, 1 h, 4 h, and 24 h) between trials [13]. 5XFAD mice showed worse performance at the 4-h retention interval than non-Tg control mice. Among the analyzed brain regions of 4-month-old 5XFAD mice, the retrosplenial cortex exhibited the highest level of Aβ plaques, followed by the frontal cortex and hippocampus [13].

Relevant neural structures are engaged by the specific demands of the learning strategy [14,15,16,17]. For example, in the water maze task, a place/spatial learning (place) strategy requiring the use of spatial cues depends critically on the hippocampus, whereas a cued/response (cued) learning strategy using egocentric navigation/instrumental learning relies on the striatum. In addition, the prefrontal cortex engages in switching from the used learning strategy to a new strategy [18,19]. Recent studies have reported that oxidative damage is observed in the frontal cortex in early AD [1,2,3], and 4-month-old 5XFAD mice may reflect the cognitive and neuropathological features of early AD. Hence, to demonstrate that oxidative damage occurs in the frontal cortex of 5XFAD mice with cognitive impairments in frontal cortex-dependent tasks, we assessed the performance of 4-month-old 5XFAD mice sequentially trained in cued and place/spatial versions of the water maze task and measured the 4-HNE levels in their frontal cortex.

## 2. Experimental Section

### 2.1. Animals

5XFAD {B6SJL-Tg (APPSwFlLon, PSEN1*M146L*L286V) 6799Vas/Mmjax} [11] Tg mice were obtained from The Jackson Laboratory (Bar Harbor, Maine, USA) and 4-month-old hemizygous 5XFAD mice (male, *n* = 12; female, *n* = 8) and non-Tg littermates (male, *n* = 9; female, *n* = 9) were used. Mice were bred and kept in a temperature (22 ± 1 °C) and humidity (50 ± 10%) controlled room with a 12-h light/dark cycle (lights on 08:00–20:00). All procedures were performed during the light cycle. Food and water were available ad libitum. The Institutional Animal Care and Use Committee of Konkuk University approved all protocols in this study (KU19037, 2 October 2019).

### 2.2. Behavioral Training Procedure

Each mouse serially received a cued/response and place/spatial version of the water maze task, adopted from McDonald and White [14] and modified in our previous studies [17,19,20,21]. Briefly, each mouse received cued/response training for 4 days, followed by place/spatial training for four days. On day 9, a competition test was performed. Before starting the task, each mouse was moved into the testing room and placed in the holding cage. During the task, mice were placed into the water facing the wall and allowed to swim for a maximum of 60 s. A trial ended when a mouse climbed onto the available platform, or after 60 s had elapsed. If the mouse did not find the platform during a trial, it was placed on the platform with the experimenter’s guidance. The mouse was left on the platform for 20 s and then moved to a holding cage for a 10-min intertrial interval.

#### 2.2.1. Apparatus

The water maze consisted of a circular tank (1.83 m diameter, 0.58 m height) with an escape platform (20 cm diameter) centered in one of the four maze quadrants. The water was made opaque using non-toxic white paint. Black sandpaper was attached to the visible escape platform to help the mouse climb. The visible platform was raised 2 cm above the water surface for cued training. The hidden platform was located 0.5 cm beneath the surface for place training. The maze was surrounded by white curtains, on which black felt patterns were affixed to provide distal visual cues. Data were recorded using an HVS Image tracking system (Water 2100, HVS Image, Hampton, UK).

#### 2.2.2. Cued/Response Training

Mice underwent four trials/day (10-min intertrial interval, maximum trial duration of 60 s, with 20 s on the platform at the end of each trial) in which a visible platform was moved to different locations in the pool between trials. Blank white curtains were drawn around the pool to conceal distal visual cues.

#### 2.2.3. Place/Spatial Training

Mice underwent four trials/day (10-min intertrial interval, maximum trial duration of 60 s, with 20 s on the platform at the end of each trial), with each trial beginning at one of four equidistantly located positions at the perimeter of the maze. The location of the platform remained constant across all training trials.

#### 2.2.4. Test for the Preference of Learning Strategy

On day 9, a competition test was held in which the visible platform was positioned in the NW quadrant (opposite the position of a hidden platform on previous place/spatial training days). Two trials were given with start points equidistant from the two platform locations (NW and SE). Swimming patterns of each mouse were analyzed to determine if mice swam to the previous hidden platform location before escaping to the visible platform. The classification of strategy preference was adopted from McDonald and White [14]. Briefly, if a mouse visited the location where the platform had been on the previous place/spatial training days, the mouse was classified as using a spatial strategy. Otherwise, the mouse was considered to be using a cued strategy.

### 2.3. Brain Preparation

Each mouse that participated in the behavioral task was sacrificed 7 days following the competition test [20,22]. Frontal cortices, hippocampal structures, and striatum were rapidly dissected and frozen at −70 °C until further processing. A separate cohort of mice was sacrificed using intracardiac perfusion. All mice were deeply anesthetized with 5% isoflurane, followed by perfusion with 0.01 M phosphate-buffered saline (PBS), and then 4% paraformaldehyde (PFA) in 0.01 M PBS. The brain was removed and post-fixed for 2 days in 4% PFA in 0.01 M PBS and then in 30% sucrose in 0.01 M PBS until the brain sank to the bottom of the container. The brain was frozen with powdered dry ice and stored at −80 °C until sectioning. The perfused brains were sectioned coronally, at a thickness of 30 μm using a microtome. The free-floating sections were stored at −20 °C in cryoprotectant (30% ethylene glycol, 25% glycerol, 25% 0.1 M phosphate buffer, and 20% distilled water).

### 2.4. Immunoblot Analyses

The brain structures were homogenized in cold lysis buffer (20 mM Tris/HCl, pH 7.5, 1.5 mM EDTA, 1% Triton X-100, 5% glycerol, 40 mM KCl, 1 mM Na_3_VO_4_, 1 mM PMSF, 0.5 mM DTT, and 1 mM NaF) [22]. After centrifugation at 14,000 rpm for 60 min at 4 °C, protein concentrations in the supernatants were measured using a bicinchoninic acid assay kit (Sigma-Aldrich, St. Louis, MO, USA). The samples were loaded (40 μg) and separated using electrophoresis in an 8% sodium dodecyl sulfate-polyacrylamide gel and transferred to a PVDF membrane. Next, membranes were stained with Ponceaus S solution (0.1% [*w*/*v*] Ponceau S in 5% [*v*/*v*] acetic acid) to verify equal loading and transfer of proteins. The membranes were blocked with 5% (*w*/*v*) skim milk for an hour, and then incubated overnight at 4 °C with rabbit anti-4-HNE (1:1000, Abcam, Cambridge, UK). Membranes were then incubated with HRP-conjugated secondary antibody solutions (goat anti-rabbit, 1:2000, Cell Signaling Technology, Danvers, MA, USA) for an hour; 1× Signal Bright Enhancer (Jubiotech, Daejeon, Korea) was used during each antibody incubation step. Detection was performed using an Amersham ECL Prime Western Blotting Detection Reagents (GE Healthcare, Chicago, IL, USA) and visualized using the ImageQuant LAS 500 (GE Healthcare). Each visualized image was analyzed using ImageJ software (Image J 1.51j8, NIH, Bethesda, Rockville, MD, USA).

### 2.5. Immunofluorescence Staining

The brain sections were washed with 0.01 M PBS, containing 0.3% Triton-X100 (PBST). The washed sections were incubated with 10 mM sodium citrate buffer for 30 min at 85 °C for antigen retrieval and then were cooled down for 30 min at room temperature.

After antigen retrieval, the brain sections were rewashed with PBST and incubated for 3 h, at room temperature, in PBST containing 10% horse serum to block non-specific reactions. Sections were incubated overnight at room temperature in a primary antibody solution cocktail, which contained rabbit anti-4-HNE (1:200, Abcam), mouse anti-4G8 (1:1000, Covance, Princeton, NJ, USA), and guinea pig anti-NeuN (1:1000, Millipore, Burlington, MA, USA). The sections were then incubated for 2 h at room temperature in a secondary antibody solution cocktail, which contained AlexaFluor 488 donkey anti-rabbit antibody (1:200, Invitrogen, Waltham, MA, USA), AlexaFluor 568 donkey anti-mouse antibody (1:200, Invitrogen), and AlexaFluor 633 goat anti-guinea pig antibody (1:200, Invitrogen). Following the incubation, the sections were mounted onto resin-coated slides and coverslipped using Prolong diamond^®^ anti-fade reagent (Invitrogen).

The stained sections were assessed, and microscopic photographs were taken using a confocal microscope (LSM900, Zeiss, Oberkochen, Germany). The microscopic photographs were analyzed using Zen 3.0 blue edition software (Zeiss). The intensity of 4-HNE, 4G8, and NeuN-positive signals were measured in the following regions of interest (ROIs): frontal cortex area 2 (Fr2, bregma 2.1 mm to 1.7 mm), medial prefrontal cortex (mPFC, bregma 2.1 mm to 1.7 mm), striatum (STR, bregma 1.18 mm to 0.62 mm), and hippocampus (HC, bregma −1.2 mm to −2.0 mm). Three sections of the frontal cortices, two sections of the striatum, and three sections of the hippocampus were used in analyses. The mean immunoreactive signal intensities for each animal within a group were averaged to obtain a mean. All immunoreactive signal intensities of neurons in the corresponding ROI were obtained by subtracting the signal intensities in areas where the neurons were not placed.

### 2.6. Statistical Analyses

The distance of the mouse from the escape platform was sampled 10 times per sec, and these values were averaged in 1-sec bins during each trial. Cumulative search error was then calculated as the summed 1-sec averages of this proximity measure corrected for the particular start location on each trial [23]. Cumulative search errors during cued and place training were separately analyzed using two-way repeated-measures analysis of variance (ANOVA; group × trial session [day]) to assess learning acquisition. For the analysis of the competition test, χ^2^ analysis was conducted to evaluate which strategy the subjects preferred, either cued/response or place/spatial strategy. During the place/spatial training phase, each mouse’s search pattern was analyzed according to Janus’ report [24]. Student’s *t*-test was used to analyze the percentage of spatial strategy each mouse adopted and the levels of 4-HNE-modified protein. Mann-Whitney U test was conducted to analyze the immunofluorescent intensity of 4-HNE and 4G8-positive signal because the assumption of homogeneity of variance was violated. Data presented in the bar graph are expressed as mean ± standard error of the mean (SEM).

## 3. Results

### 3.1. 4-Month-Old 5XFAD Mice Use Less Spatial Strategy than Non-Tg Mice in Place Learning Following Cued Learning

Figure 1A shows the cumulative search errors of 4-month-old 5XFAD and non-Tg mice in the cued training with a visible platform and subsequent spatial training with a hidden platform. 5XFAD and non-Tg mice exhibited equal performance during the cued training. A two-way repeated-measures ANOVA on the search error of the cued training revealed the significant effect of the session (F_(3,78)_ = 29.481, *p* < 0.001), but no significant effect of group (F_(1,26)_ = 0.28, *p* = 0.601) and no significant interaction effect of group × session (F_(3,78)_ = 1.059, *p* = 0.371). A similar pattern was observed in subsequent place training. Two-way repeated-measures ANOVA on the search error of the place training revealed a significant effect of the session (F_(3,78)_ = 3.250, *p* < 0.05), but no significant effect of group (F_(1,26)_ = 2.957, *p* = 0.097) and no significant interaction effect of group × session (F_(3,78)_ = 1.339, *p* = 0.268). Procedural knowledge of the task acquired in the prior cued training has been reported to interfere with the use of the spatial strategy needed to perform the place task efficiently more in 6-month-old 5XFAD mice than non-Tg control mice [20]. Therefore, we divided place training into two sessions (the first 8 trials and second 8 trials) and analyzed the search patterns of mice using criteria mentioned earlier (Figure 1B) [19,20,24]. Student’s *t*-test on the percentage of mice that used a spatial strategy in the place training revealed that 4-month-old 5XFAD mice used this strategy less than non-Tg mice in both sessions (session 1, t_(26)_ = 2.957, *p* < 0.005; session 2, t_(26)_ = 4.170, *p* < 0.001; Figure 1B). In the competition test, 7 of 14 control mice and 6 of 14 5XFAD mice used a spatial strategy. χ^2^ analysis (χ^2^ = 0.144, *p* = 0.705) showed no significant between-group differences (Figure 1C,D).

### 3.2. Higher Levels of 4-HNE in the Prefrontal Cortex of 4-Month-Old 5XFAD Mice than Non-Tg Mice

Figure 2 shows the expression levels of 4-HNE in the frontal cortex, hippocampus, and striatum of 4-month-old 5XFAD and non-Tg control mice. We chose five bands with evident immunoreactive signals and then measured the immunoreactivities of each band. Student’s *t*-test revealed that two bands (120 and 170 kDa) in the frontal cortex were significant higher in 5XFAD mice than in control mice (120 kDa, t_(6.162)_ = −5.303, *p* < 0.005; 170 kDa, t_(11)_ = −3.332, *p* < 0.05). In the hippocampus and striatum, no differences between the two strains were found in any band.

### 3.3. Appearance of Oxidative Damage in the Frontal Cortex of 4-Month-Old 5XFAD Mice

Based on the immunoblot showing that levels of 4-HNE protein adducts increased in the frontal cortex of 5XFAD mice, we measured 4-HNE and Aβ levels, alongside neuronal status of the frontal subregions, medial prefrontal cortex (mPFC), and frontal cortex area 2 (Fr2) in the 4-month-old 5XFAD and non-Tg control mice, using triple immunofluorescence with antibodies against 4-HNE, 4G8, and NeuN (Figure 3). The Fr2 and mPFC in the 5XFAD mice showed higher 4-HNE levels than those in non-Tg mice (Mann-Whitney U test, Fr2, U = 1, *p* < 0.05; mPFC, U = 2, *p* < 0.05; Figure 3C). Aβ deposition levels were also higher in the Fr2 (U = 0, *p* < 0.05) and mPFC (U = 0, *p* < 0.05) in 5XFAD mice than those in control mice (Figure 3D). No differences in NeuN-positive immunoreactivities were observed in the Fr2 (t_(4.164)_ = −2.482, *p* = 0.066) or mPFC (t_(8)_ = −0.106, *p* = 0.918) (Figure 3E).

We examined 4-HNE and 4G8-positive signals and NeuN-positive signals in the hippocampus and striatum in 4-month-old 5XFAD mice (Figure 4). No differences between the two groups in 4-HNE levels in the hippocampus and striatum were found (Mann-Whitney U, hippocampus, U = 5, *p* = 0.17; striatum, U = 9, *p* = 0.61, Figure 4B). The intensities of 4G8-positive signals in the 5XFAD mice’s hippocampus were higher than those in the control mice (U = 0, *p* < 0.05). However, there was no difference in the signal intensities in the striatum between the two groups (U = 6, *p* = 0.26) (Figure 4C). No differences in the levels of NeuN-positive signals in the hippocampus and striatum were observed between the two groups (hippocampus, t_(8)_ = −1.560, *p* = 0.157; striatum, t_(8)_ = 0.683, *p* = 0.514, Figure 4D).

## 4. Discussion

In the AD brain, oxidative stress induced by accumulation of Aβ peptides leads to inflammation and mediates neurotoxic effects of protein aggregates, including ceramide, resulting in functional impairments [25]. Cognitive decline in AD is more evident, with more severe lipid peroxidation indexed by protein-bound 4-HNE [7]. Oxidative damage is observed in the frontal cortex of patients with early AD [6,7]. Interestingly, oxidative damage is more pronounced in the frontal cortex than in other cortices in the AD brain [9,10]. Hence, to shed more light on the molecular/cellular processes of early AD pathogenesis and develop therapeutics for AD, an animal AD model with the cognitive and pathological characteristics of early AD is needed. The 5XFAD mouse is one of the most used animal AD models, and recent studies suggest that 4-month-old 5XFAD mice may reflect early AD pathologies [12,26]. Therefore, we first assessed cognitive flexibility to examine frontal cortex-related cognitive function in 4-month-old 5XFAD mice using a learning strategy-switching task from a cued to spatial strategy in a water maze. 5XFAD mice exhibited less use of the spatial strategy in the switched place training than non-Tg mice. To evaluate the status of oxidative damage in the frontal cortex of non-Tg and 5XFAD mice, we measured the immunoreactivities of 4-HNE protein, a marker of oxidative stress, in the frontal cortex, hippocampus, and striatum. The 4-HNE levels were higher in the frontal cortex in 5XFAD mice than in non-Tg mice.

The frontal cortex responsible for executive and higher cognitive functions, such as cognitive flexibility, is one of the most affected brain regions in the AD brain [27,28]. Interestingly, more significant oxidative damage is observed in the frontal cortex than other brain regions in AD brain [9,10]. Research on animal AD models reported the same results. For example, Girard et al., (2013) investigated frontal cortex integrity of 2-, 4-, and 6-month-old 5XFAD mice using the olfactory H-maze task developed for examining the frontal cortex. Behavioral impairments first were appeared in 4-month-old 5XFAD mice. In addition, 4-month-old 5XFAD mice showed heavy gliosis and amyloid deposition in the frontal cortex. Working memory traditionally reported to be related to the frontal cortex was also evaluated in 4-month-old 5XFAD mice using the novel object recognition task. 4-month-old 5XFAD mice showed worse performance at the 4-h retention interval than non-Tg control mice [13].

The frontal cortex, including the prefrontal cortex, is also known to be critically engaged in strategy switching [19,29,30]. We previously reported that 3-month-old C57BL/6 mice with lesions of medial prefrontal cortex or orbitofrontal cortex exhibited continued use of the cued search strategy acquired during cued training in the switched place training phase [19]. We also examined 6-month-old 5XFAD mice’s performance in the same learning strategy-switched task and observed the same behavioral pattern as that seen in mice with prefrontal cortex lesions. Subsequently, the present study found cognitive impairments in the learning strategy-switching task supporting that 4-month-old 5XFAD mice may be a suitable animal model for early AD [12]. Specifically, 4-month-old 5XFAD and non-Tg mice showed equal performance during both cued and place training phases and competition tests. However, 4-month-old 5XFAD mice continued to use the cued search strategy during the place training following the cued training. These results indicate that 5XFAD mice are not proficient in switching search strategies.

Most studies of 5XFAD mice have investigated Aβ deposition in brain regions, including the hippocampus which is responsible for memory impairments. Our quantitative analysis of regional Aβ42 deposition in 6-month-old 5XFAD mice using ELISA revealed that levels were highest in the hippocampus, followed by the cortex, then the striatum [19]. The present analysis of regional Aβ42 deposition in 4-month-old 5XFAD mice using immunofluorescence staining exhibited differences between the 5XFAD and non-Tg mice in the frontal cortex and hippocampus, but not in the striatum. We further examined oxidative damage in the several brain regions of 4-month-old 5XFAD mice by measuring 4-HNE levels using immunoblot. As a result, significantly higher amounts of 4-HNE proteins were detected in the frontal cortex of 4-month-old 5XFAD mice, whereas the hippocampus and the striatum did not show any significant differences between 5XFAD and control mice. These results indicate that the frontal cortex of 4-month-old 5XFAD mice is the first region to become impaired. However, the oxidative damage in the frontal cortex of these mice is needed to be confirmed by measuring the other oxidative stress markers and antioxidants.

Examination of oxidative stress in brain cortices isolated from 3- to 5-month-old triple-transgenic AD mice showed decreased levels of nonenzymatic antioxidants and increased expression levels of lipid peroxidation markers [31]. In a human postmortem study, a significant decline in antioxidants was observed in the frontal cortex of patients with MCI and both early- and late-stage AD [32]. Considering these reports and the present findings that 4-month-old 5XFAD mice showed impairments in the learning strategy-switching tasks requiring the frontal cortex as well as high amounts of oxidatively damaged proteins in their frontal cortex, 4-month-old 5XFAD mouse is a suitable animal model for early AD.

## Figures and Tables

**Figure 1 biomedicines-08-00326-f001:**
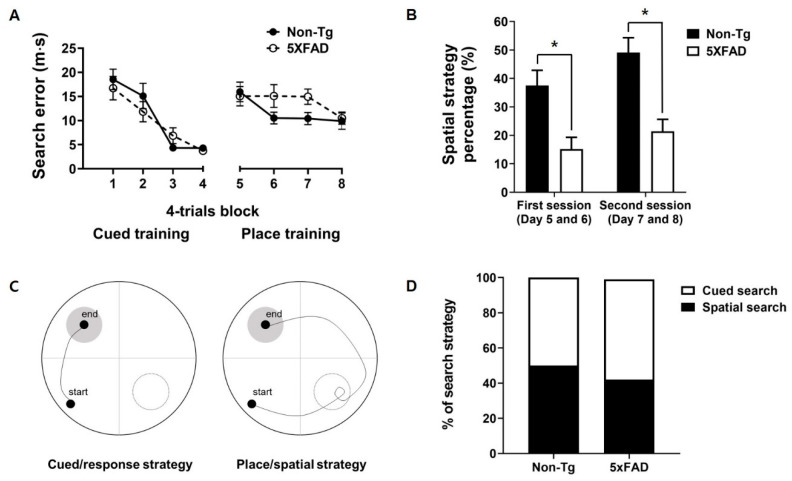
Performance of 4-month-old 5XFAD and non-Tg control mice in the switching task from cued/response training to place/spatial training. (**A**) No significant difference between non-Tg and 5XFAD mice in search error was observed in the cued/response training and place/spatial training. (**B**) 5XFAD mice adopted the spatial search strategy less in the place/spatial training than did control mice. (**C**) Representative figure of the swim path in the competition test. (**D**) No difference between 5XFAD and control mice in the preference for a search strategy in the competition test was found. Non-Tg, *n* = 14; 5XFAD, *n* = 14. * *p* < 0.05.

**Figure 2 biomedicines-08-00326-f002:**
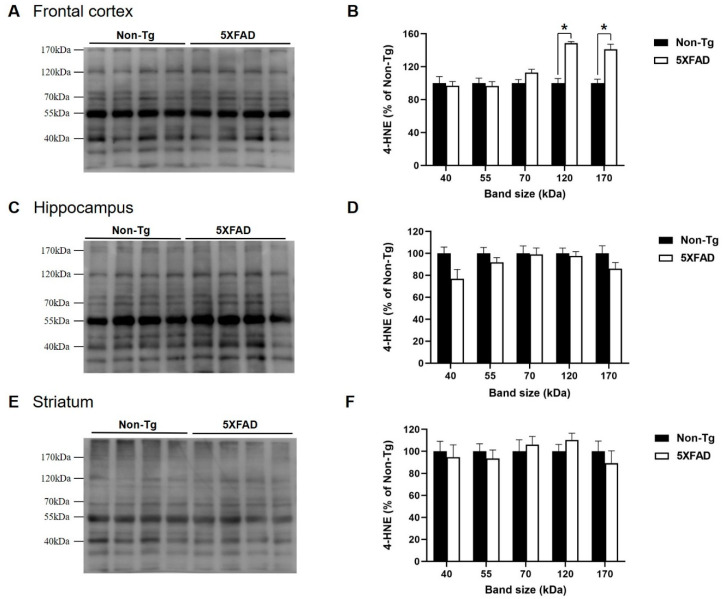
Increased levels of 4-HNE protein adducts in the frontal cortex of 4-month-old 5XFAD mice. Representative immunoblots of the frontal cortex (**A**), hippocampus (**C**), and striatum (**E**) of 4-month-old 5XFAD and non-Tg control mice. Five bands with evident immunoreactivity (40, 55, 70, 120, and 170 kDa) were selected and normalized according to 55 kDa band of Ponceau S staining. 4-HNE levels in the frontal cortex (**B**), hippocampus (**D**), and striatum (**F**) of 5XFAD mice represented as a percentage of the levels in non-Tg control mice. (**B**) 4-HNE levels in the frontal cortex of 5XFAD mice were significantly higher than those in non-Tg mice in specific band sizes (120 and 170 kDa). In the hippocampus (**D**) and striatum (**F**), no significant differences between the two mice were observed in any band size. Non-Tg, *n* = 6–8; 5XFAD, *n* = 6–8; * *p* < 0.05.

**Figure 3 biomedicines-08-00326-f003:**
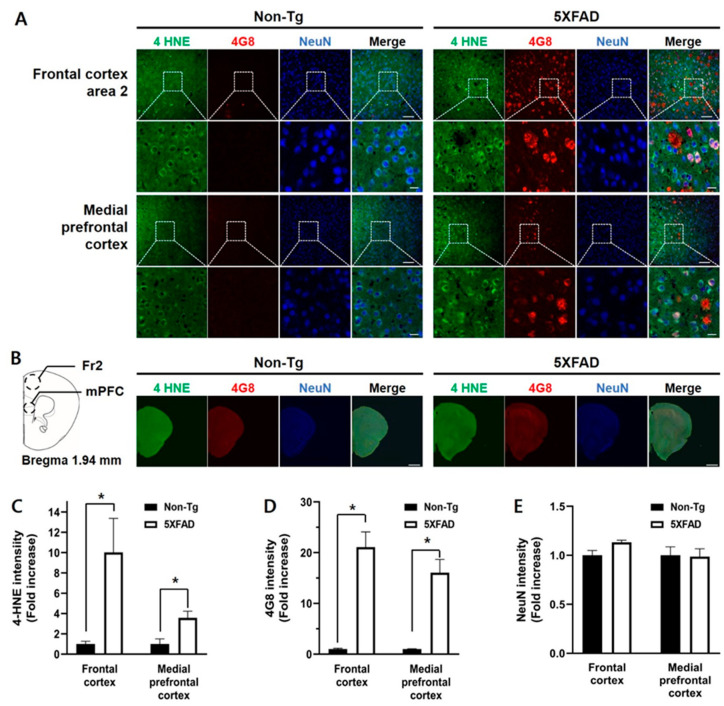
Triple immunofluorescence staining of the frontal cortex of the 4-month-old non-Tg control and 5XFAD mice. (**A**) Confocal microscopic images of triple immunofluorescence with 4-HNE, 4G8, and NeuN antibodies in the frontal cortex area 2 (Fr2) and medial prefrontal cortex (mPFC). (**B**) Schematic diagram of the analyzed brain regions and half-brain immunofluorescence images. The dashed circles in the diagram indicate the regions of interest. (**C**) 4-HNE levels in the Fr2 and mPFC in 5XFAD mice were significantly higher than those in control mice. (**D**) Aβ levels in the Fr2 and mPFC in 5XFAD mice were significantly higher than those in control mice. (**E**) There was no difference in the intensities of the NeuN-positive signal. Top row: scale bar indicates 100 μm; bottom row: scale bar indicates 20 μm. Non-Tg, *n* = 4; 5XFAD, *n* = 6. * *p* < 0.05.

**Figure 4 biomedicines-08-00326-f004:**
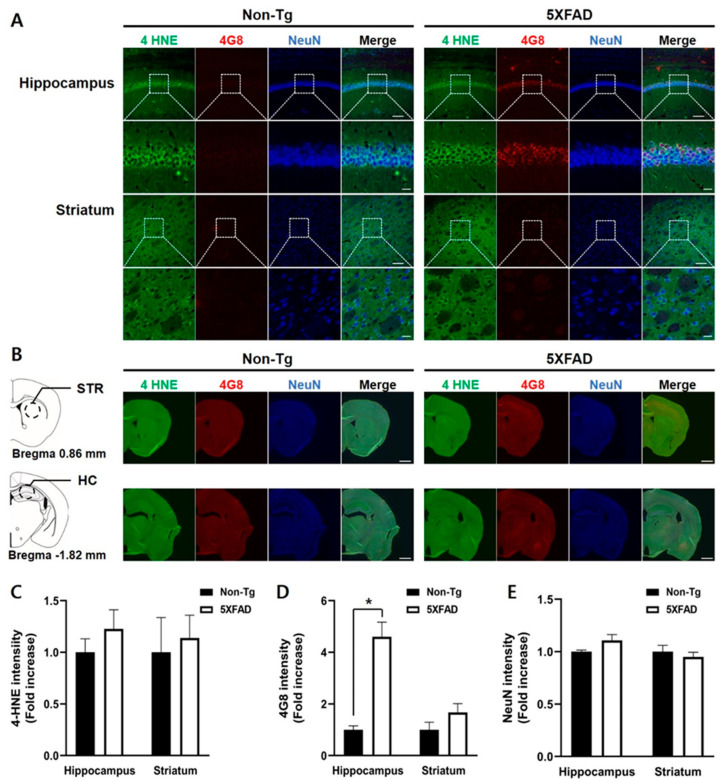
Triple immunofluorescence staining of the hippocampus and striatum of the 4-month-old non-Tg control and 5XFAD mice. (**A**) Confocal microscopic images of triple immunofluorescence with 4-HNE, 4G8, and NeuN antibodies in the hippocampus (HC) and striatum (STR). (**B**) Schematic diagram of the analyzed brain regions and half-brain immunofluorescence images. The dashed circle line in the diagrams indicates the region of the interest. (**C**) No differences between the two groups in the 4-HNE levels of the hippocampus and striatum were found. (**D**) Levels of 4G8-positive signals were significantly higher in the hippocampus of 5XFAD than that in control mice, but not in the striatum. (**E**) No differences in intensities of NeuN-positive signals were found between the two groups. Top row: scale bar indicates 100 μm; bottom row: scale bar indicates 20 μm. Non-Tg, *n* = 4; 5XFAD, *n* = 6. * *p* < 0.05.

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
