# Peer review of "4-Hydroxynonenal Immunoreactivity Is Increased in the Frontal Cortex of 5XFAD Transgenic Mice"

_biomedicines, 2020, doi:10.3390/biomedicines8090326_

Round 1

Reviewer 1 Report

In this study, the authors used 4-month-old five familial Alzheimer’s disease (AD; 5XFAD) transgenic mice to evaluate them as a potential model of early AD pathology.

They assessed changes in behavior dependent on hippocampus, striatum and frontal cortex by using several Morris water maze protocols, as oxidative damage in the same structures by using 4-hydroxy-2-trans-nonenal (4-HNE) as a marker of oxidative stress. The methodology is sound, and the results are interesting, indicating increased 4-HNE / oxidative stress in the frontal cortex / PFC, and a corresponding impairment in strategy shifting /behavioral flexibility. The following points should be addressed:

  • Please specify in methods how the search error / cumulative search error was assessed
  • In the frontal cortex and striatum immunoblots, the 55 kDa band seems very saturated and therefore difficult to do a densitometric analysis. A lower concentration of protein should be loaded to specifically evaluate that band.
  • Please clarify how many animals per group were used for immunohistochemistry in the text or figure legends. It is stated in methods the number of slices (lanes 156 and 157), but the number of animals should also be stated.
  • In the immunohistochemistry panels, while a scheme of the ROI is presented, it is always better to show one full immunolabeled slice per region evaluated (and per group) so as one can have a better appreciation of the staining patterns and changes observed.
  • Still in the immunohistochemistry panels, dashed circles indicate ROIs, which are presented in higher magnification just below. However, this should be made clearer by for instance by drawing straight lines from the ROI until the magnified images, by stating that in the figure legend or by attaching a letter to each panel.
  • Does the amount of 4-HNE proteins in the frontal cortex / PFC correlate with the severity of impairment observed in the 5xFAD mice?
  • The authors used male and female animals. Gender-specific differences in pathology have been reported using the 5xFAD. The authors should clarify whether that was evaluated, and no differences were found. It is necessary to discuss how such a difference, at least reported by others, may have impacted their results.
  • The discussion is rather short. While the authors used only one behavioral test, there is also literature on prefrontal cortex changes and related behavior that should be discussed to integrate the findings reported. Better discussed should also be how 4-HNE-modified proteins correlate with disease progression and severity both in models and humans – this is stated in the introduction but not further developed in the discussion to integrate the findings.

Author Response

  • Please specify in methods how the search error / cumulative search error was assessed

We inserted the sentence regarding the search error / cumulative search error in the 2.6.

  • In the frontal cortex and striatum immunoblots, the 55 kDa band seems very saturated and therefore difficult to do a densitometric analysis. A lower concentration of protein should be loaded to specifically evaluate that band.
  • We uploaded less saturated immunoblot figures than before. We also found no significant differences between non-Tg and 5XFAD mice in the uploaded immunoreactivity (Frontal cortex, t(6)=-0.395, p=0.707; Striatum, t(6)=1.997, p=0.093).

Figure S1. Representative immunoblots of the frontal cortex (A) and striatum (C). 55 kDa bands were normalized according to 55 kDa band of Ponceau S staining. 4-HNE levels in the frontal cortex (B) and striatum (D) of 5XFAD mice represented as a percentage of non-Tg control mice. Non-Tg, n=4; 5XFAD, n=4 (Please see the attachment))

  • Please clarify how many animals per group were used for immunohistochemistry in the text or figure legends. It is stated in methods the number of slices (lanes 156 and 157), but the number of animals should also be stated.
  • Thanks for pointing it out. We put the number of animals (non-Tg control, n=4; 5XFAD, n=6) used in immunohistochemistry in figure legends.
  • In the immunohistochemistry panels, while a scheme of the ROI is presented, it is always better to show one full immunolabeled slice per region evaluated (and per group) so as one can have a better appreciation of the staining patterns and changes observed. 
  • According to the reviewer’s comments, half-brain immunofluorescence images were presented in Figure 3 and 4.
  • Still in the immunohistochemistry panels, dashed circles indicate ROIs, which are presented in higher magnification just below. However, this should be made clearer by for instance by drawing straight lines from the ROI until the magnified images, by stating that in the figure legend or by attaching a letter to each panel. 
  • According to the reviewer’s comments, the straight dashed line was inserted in Figure 3A and 4A.
  • Does the amount of 4-HNE proteins in the frontal cortex / PFC correlate with the severity of impairment observed in the 5xFAD mice? 
  • Significant behavioral difference between 5XFAD and non-Tg mice was the presence/absence of the spatial strategy choice over the place/spatial training following the cued/response learning. In essence, these were the nominal variables. Even though two variables are ratio, a correlation with a small size of the subject would lead us to an inflected conclusion. In the present study, the number of animals per each group was 6-8 mice.
  • The authors used male and female animals. Gender-specific differences in pathology have been reported using the 5xFAD. The authors should clarify whether that was evaluated, and no differences were found. It is necessary to discuss how such a difference, at least reported by others, may have impacted their results.The sex difference is one of the controversial research topics. The behavioral tasks requiring an extended training duration (the mice were trained for nine days in the present study) should be conducted considering a mouse estrous cycle (4-5 days) and animal age [1]. For example, spatial learning in the Morris water maze was equivalent in the 6-month-old male and female rats; younger animals reported sex differences [2].
  • We conducted a two-way repeated-measures ANOVA (group: male vs female, only 5XFAD mice) on the search error of the cued training (Session, F(3,36)=13.87, p<0.001; Group × Session (F(3,36)=3.186, p<0.05; Group, F(1,12)=1.509, p=0.243). Student’s t-test on the percentage of a spatial strategy in the place training revealed that male 5XFAD mice used this strategy less than female mice in the first session, but not in the second session. In the competition test, 5 of 8 male mice and 1 of 6 female mice used a spatial strategy. χ2 analysis (χ2=2.941, p=0.086) revealed no significant sex difference.
  •  

Figure S2. Performance of 5XFAD male and female mice in the switching task from cued/response training to place/spatial training. (A) No significant difference was found between male and female mice in the search error during both training. (B) During the place training, female 5XFAD mice used spatial strategy more often than male 5XFAD mice in the first session, but not in the second session (C) No sex differences were found in the preference for a search strategy in the competition test. male, n=8; female, n=6. *p<0.05.

In the immunofluorescent and immunoblot analysis, The Mann-Whitney U test was conducted because the assumption of homogeneity of variance was violated. No sex differences were found in the immunofluorescent and immunoblot analysis.

Frontal cortex

Medial prefrontal cortex

Hippocampus

Striatum

4-HNE

U=7, p=0.267

U=5, p=1

U=8, p=0.133

U=8, p=0.133

4G8

U=4, p=1

U=5, p=1

U=4, p=1

U=2, p=0.533

NeuN

U=2, p=0.533

U=2, p=0.533

U=4, p=1

U=8, p=0.133

Table 1. Mann-Whitney U test statistics for immunofluorescence analysis. male, n=4; female, n=2

Frontal cortex

Hippocampus

Striatum

40kDa

U=2, p=0.857

U=13, p=0.2

U=7, p=0.267

55kDa

U=3, p=1

U=9, p=1

U=8, p=0.133

70kDa

U=2, p=0.857

U=12, p=0.343

U=4, p=1.000

120kDa

U=4, p=1

U=13, p=0.2

U=7, p=0.267

170kDa

U=1, p=0571

U=10, p=0.686

U=8, p=0.133

Table 2. Mann-Whitney U test statistics for immunoblot analysis. frontal cortex, male, n=6, female, n=1; hippocampus, male, n=4, female, n=4; striatum, male, n=4, female, n=2.

Sex differences in 5XFAD mice were variable in the present study. Some studies have reported sex differences in 5XFAD mice; the other studies have reported no significant sex differences in 5XFAD mice. For example, 5XFAD mice did not show sex differences in the beta-amyloidogenic pathway [3, 4] and object recognition memory deficits [5]. The present study’s purpose was the suitability of 4-month-old 5XFAD as an early AD model, not on the sex differences. To reveal sex differences in 5XFAD mice in strategy switching, we have to arrange the other experiment with n larger size.

The discussion is rather short. While the authors used only one behavioral test, there is also literature on prefrontal cortex changes and related behavior that should be discussed to integrate the findings reported. Better discussed should also be how 4-HNE-modified proteins correlate with disease progression and severity both in models and humans – this is stated in the introduction but not further developed in the discussion to integrate the findings.

According to the reviewer’s comments, we revised the discussion.

Reviewer 2 Report

Main comments
Why the authors did not assess the expression of 4-HNE protein adducts? Free 4-HNE is characterized by completely different properties.

2.1 Why control is not equal to the study group? Why a males is not as much as females?
2.2. Is the 4-day training sufficient? Give adequate citation.
2.3 How was the time of tissue collection established? What about other brain structures?
Add citations to the methodology.
2.4 Add citations to the methodology.
3) Why the authors do not present data with a distinction by gender? Why were the experiments carried out on females as well and not on males as usual?
4) This part of the manuscript is too laconic and does not lead to specific conclusions. What is the importance of research? Furthermore, discuss briefly the relationship between oxidative damage, inflammation and ceramide production in AD (see: 10.3390/ijms20040874).
What are the limitations of the study? What is the next step?

Author Response

Please see the attachment for our responses to the other reviewer's comment and the references.

Why the authors did not assess the expression of 4-HNE protein adducts? Free 4-HNE is characterized by completely different properties.

Thank the reviewer for pointing it out. We used rabbit polyclonal 4-HNE antibody to detect 4-HNE bound proteins. The antibody binds to the 4-HNE adducts. We described as 4-HNE protein adducts in Figure 2 and 3.3.

2.1 Why control is not equal to the study group? Why a males is not as much as females?

We generated these mice here. The number of these mice were controlled by the strict regulation of the Institutional Animal Care and Use Committee of Konkuk University. Furthermore, we have to report the number of animals used in the experiment and use as few mice as possible. We tried to have the same number of animals per group, including males and females, at the beginning of the experiment. We lost a few animals over the training course because of thigmotaxis and weight loss. We kept trying to have an equal number per group in the next run, but we could not.

2.2. Is the 4-day training sufficient? Give adequate citation.

- Our training protocol consist of cued/response training for 4 days and subsequent spatial/place training for 4 days. This training protocol has been used in our previous studies [6-9]. We citated these papers in the method.

2.3 How was the time of tissue collection established? What about other brain structures?
Add citations to the methodology.

We have published a couple of papers with the protocol [7, 10]. We cited these papers.

2.4 Add citations to the methodology.

We cited the paper regarding this methodology.

3) Why the authors do not present data with a distinction by gender? Why were the experiments carried out on females as well and not on males as usual?

As we answered the other reviewer’s comment above, sex differences in 5XFAD mice were variable in the present study. Several studies have reported sex differences in 5XFAD mice; the other studies have reported no significant sex differences in 5XFAD mice. For example, 5XFAD mice did not show sex differences in the beta-amyloidogenic pathway [3, 4] and object recognition memory deficits [5]. The present study’s purpose was the suitability of 4-month-old 5XFAD as an early AD model, not on the sex differences. To reveal sex differences in 5XFAD mice in strategy switching, we have to arrange the other experiment with n larger size.

We mentioned the sex-related issue earlier.

4) This part of the manuscript is too laconic and does not lead to specific conclusions. What is the importance of research? Furthermore, discuss briefly the relationship between oxidative damage, inflammation and ceramide production in AD (see: 10.3390/ijms20040874). 
What are the limitations of the study? What is the next step?

According to the reviewer’s comments, we revised the discussion. We focused on the suitability of 4-month-old 5XFAD as an early AD model. We have shown that 4-month-old 5XFAD mice exhibited impairments in the frontal cortex-related behavioral task and high amount of oxidatively damaged proteins in the frontal cortex. MCI and early AD patients showed impairments in the frontal cortex-related cognitive function and elevated 4-HNE modified proteins in their frontal cortex. Therefore, we suggest 4-month-old 5XFAD mice as an early AD model. We will try to confirm the oxidative damage of the frontal cortex by measuring the other oxidative stress markers, and antioxidants, which was described in the discussion.

Round 2

Reviewer 2 Report

Ready for publication